# MAXIMALLY EXPRESSIVE GNNS FOR OUTERPLANAR GRAPHS

## ABSTRACT

We propose a *linear time* graph transformation that enables the Weisfeiler-Leman (WL) test and message passing graph neural networks (MPNNs) to be maximally expressive on *outerplanar* graphs. Our approach is motivated by the fact that most pharmaceutical molecules correspond to outerplanar graphs. Existing research predominantly enhances the expressivity of graph neural networks without specific graph families in mind. This often leads to methods that are impractical due to their computational complexity. In contrast, the restriction to outerplanar graphs enables us to encode the Hamiltonian cycle of each biconnected component in linear time. As the main contribution of the paper we prove that our method achieves maximum expressivity on outerplanar graphs. Experiments confirm that our graph transformation improves the predictive performance of MPNNs on molecular benchmark datasets at negligible computational overhead.

## 1 INTRODUCTION

We study graph neural networks (GNNs) for the family of outerplanar graphs and devise a model that can distinguish all non-isomorphic outerplanar graphs after applying a linear time pre-processing step. Morris et al. (2019) and Xu et al. (2019) showed that message passing graph neural networks (MPNNs) have limited *expressivity*, i.e., there exist non-isomorphic graphs on which every MPNN will produce the same embedding. Such graphs exist even within the restricted class of outerplanar graphs (see Figure 5). This led to the development of GNNs that are more expressive than MPNNs which are often called *higher-order* GNNs. However, the increase in expressivity often comes with a significant increase in computational complexity which makes these methods impractical for large graphs. For example, $k$-GNNs (Morris et al., 2019) have a time complexity of $\Omega(|V|^3)$, while other higher-order GNNs count pattern graphs such as cliques (Bodnar et al., 2021b), cycles (Bodnar et al., 2021a;b), and general subgraphs (Bouritsas et al., 2022), which can take exponential time in the pattern size. However, for certain domains of interest the graph structure can be exploited to build efficient higher-order GNNs. In this work, we focus on the pharmaceutical domain and on graphs that represent molecules. Over $92\%$ to $97\%$ of the graphs in widely used benchmark datasets in this domain are *outerplanar* (see Table 1). The properties of outerplanar graphs have been exploited by algorithms for graph mining (Horváth et al., 2010) and molecular similarity computation (Schietgat et al., 2013; Droschinsky et al., 2017). However, no efficient GNNs with expressivity guarantees on outerplanar graphs have been proposed. We focus on this class of graphs and devise a linear time transformation that allows an MPNN to become maximally expressive on outerplanar graphs.

We propose to decompose the outerplanar graphs into biconnected outerplanar components and trees. Using the fact that each biconnected outerplanar component has a unique Hamiltonian cycle that can be computed in linear time, we split each component into the two directions of the Hamiltonian cycle and prove that MPNNs are maximally expressive on biconnected outerplanar graphs transformed in this way. Taking advantage of the well-known fact that MPNNs are maximally expressive on labeled trees (Arvind et al., 2015; Kiefer, 2020), we extend our result into a pre-processing transformation called *Cyclic Adjacency Transform* (CAT) that works on all outerplanar graphs. We benchmark CAT with common MPNNs on a variety of molecular graph benchmarks and show that CAT consistently boosts the performance of MPNNs.

**Main contributions.** We propose CAT, a linear time pre-processing that renders MPNNs maximally expressive on outerplanar graphs. We prove that as a result of our transformation CAT*,

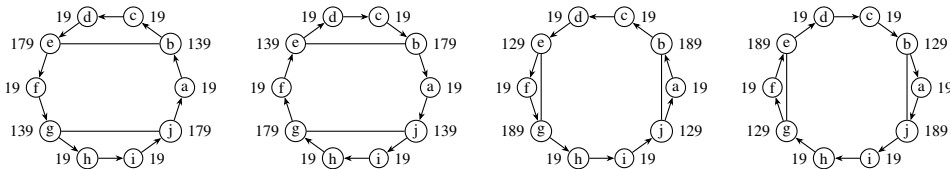

Figure 1: Two graphs and their directed Hamiltonian cycles. Nodes are annotated with their HALs, the distances on the Hamiltonian cycle to their neighbors (Colbourn & Booth, 1981).

1-Weisfeiler-Leman is maximally expressive on biconnected outerplanar graphs. We exploit the fact that outerplanar graphs can be decomposed into biconnected outerplanar components and trees: we define CAT by applying CAT* to each biconnected component and prove its maximal expressivity on outerplanar graphs.

## 2 PRELIMINARIES

A *graph* $G = (V, E, \mu, \nu)$ consists of a set of nodes $V$, a set of edges $E \subseteq V \times V$ and attributes for the nodes $\mu \colon V \to X$ and edges $\nu \colon E \to X$, respectively, where $X$ is a set of attributes. We refer to an edge from $u$ to $v$ by $uv$, and in case of undirected graphs $uv = vu$. The *in-neighbors* of a node $u \in V$ are denoted by $N_-(u) = \{v \mid vu \in E\}$. The *out-neighbors* of a node $u \in V$ are denoted by $N_+(u) = \{v \mid uv \in E\}$ and in case of undirected graphs, $N_- = N_+$. In this paper, the input graphs are undirected and are transformed into directed ones. A graph $G' = (V', E', \mu', \nu')$ is a subgraph of a graph $G$, denoted by $G' \subseteq G$, iff $V' \subseteq V$, $E' \subseteq E$, $\forall v \in V' \colon \mu'(v) = \mu(v)$, and $\forall e \in E' \colon \nu'(e) = \nu(e)$. A (directed) cycle $(v_1, \dots, v_k)$ is a sequence of $k \geq 3$ distinct nodes, with $\forall i \in \{1, \dots, k-1\} \colon v_i v_{i+1} \in E$ and $v_k v_1 \in E$. A graph is *acyclic*, if it does not contain a cycle. Given a graph $G$, we denote the shortest path distance between two nodes $u$ and $v$ by $d_G(u, v)$, or $d(u, v)$ if it is clear from the context. We denote the *diameter* of a graph $G$ by $\Phi(G) = \max_{u,v \in V(G)} d(u, v)$.

A graph is *outerplanar* if it can be drawn in the plane without edge crossings and with all nodes belonging to the exterior face (see Felsner (2012) for more details). We call an undirected graph with at least three vertices *biconnected* if the removal of any single node does not disconnect the graph. A *biconnected component* is a maximal biconnected subgraph. We refer to the outerplanar biconnected components of a graph as *blocks*.

Two graphs $G$ and $H$ are isomorphic, if there exists a bijection $\psi \colon V(G) \to V(H)$, so that $\forall u, v \in V(G) \colon \mu(v) = \mu(\psi(v)) \land uv \in E(G) \Leftrightarrow \psi(u)\psi(v) \in E(H) \land \forall uv \in E(G) \colon \nu(uv) = \nu(\psi(u)\psi(v))$. We call $\psi$ an isomorphism between $G$ and $H$. An *in-tree* $T$ is a connected, directed, acyclic graph with a distinct *root* with no outgoing edges and other nodes have one outgoing edge.

**Weisfeiler-Leman.** The 1-dimensional Weisfeiler-Leman algorithm (WL) iteratively assigns colors to nodes. The color of a node $v \in V(G)$ is updated iteratively according to $c_{i+1}(v) = h\left(c_i(v), \{\!\!\{(\nu(uv), c_i(u)) \mid u \in N_-(v)\}\!\!\}\right)$, where $h$ is an injective function and $c_0 = \mu$.

The *unfolding tree* with height $i$ of a node $v \in V(G)$ is defined as the in-tree $F_i^v = (v, \{\!\!\{F_{i-1}^u \mid u \in N_-(v)\}\!\!\})$, where $F_0^v = (\{v\}, \emptyset)$. The unfolding trees $F_i^v$ and $F_i^w$ of two nodes $v$ and $w$ are isomorphic iff the colors of the nodes in iteration $i$ are equal. For more details and a full proof see, e.g., Kriege (2022). The Weisfeiler-Leman algorithm has historically been used as a heuristic for graph isomorphism. Let $\mathrm{WL}(G) = \{\!\!\{c_\infty(v) \mid v \in V(G)\}\!\!\}$ be the multiset of node colors in the stable coloring (Arvind et al., 2015). Two graphs $G$ and $H$ are not isomorphic, if $\mathrm{WL}(G) \neq \mathrm{WL}(H)$. However, non-isomorphic graphs $G$ and $H$ with $\mathrm{WL}(G) = \mathrm{WL}(H)$ exist. WL for example cannot distinguish the molecular graphs in Figure 5 or a 6-cycle from two triangles.

**Hamilton adjacency lists.** A Hamiltonian cycle of a graph is a cycle containing each node exactly once. Biconnected outerplanar graphs have a unique Hamiltonian cycle that can be found in linear time (Mitchell, 1979). Annotating each node with the sorted distances $d_C$ to all its neighbors on the two directed variants of the Hamiltonian cycle $C$ gives us Hamiltonian adjacency lists (HALs). Figure 1 shows two graphs annotated with their HALs in both directions of the Hamiltonian cycle.

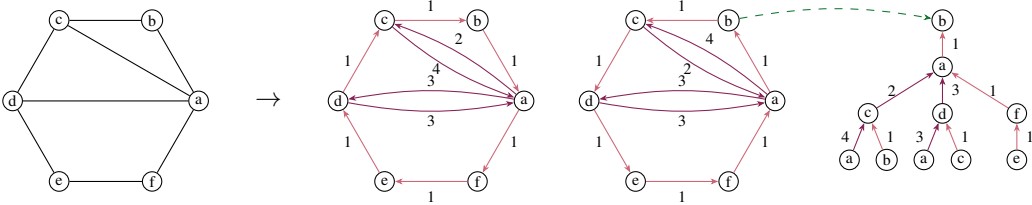

Figure 2: Biconnected outerplanar graph $G$, $\text{CAT}^*(G)$ and the unfolding tree of one of its nodes.

Table 1: Common benchmark datasets and percentage of outerplanar graphs in them.

| Dataset | #Graphs | Outerplanar |
|---|---|---|
| ZINC | 12000 | 98 % |
| PCQM-Contact | 529434 | 98 % |
| MOLESOL | 1128 | 97 % |
| MOLTOXCAST | 8576 | 96 % |
| MOLTOX21 | 7831 | 96 % |
| MOLLIPO | 4200 | 96 % |
| MOLCLINTOX | 1477 | 94 % |
| NCI-2000 | 250251 | 94 % |
| peptides-func | 15535 | 93 % |
| MOLBACE | 1513 | 93 % |
| MOLSIDER | 1427 | 92 % |
| MOLBBBP | 2039 | 92 % |
| MOLHIV | 41127 | 92 % |

Table 2: Pre-processing time of CAT on the training splits of all datasets and relative additional training / evaluation time with CAT.

| Dataset | CAT Runtime | Tra+Eva Time Increase |
|---|---|---|
| MOLESOL | $2 \pm 1$ s | 26 % |
| MOLBBBP | $5 \pm 1$ s | 36 % |
| MOLSIDER | $6 \pm 1$ s | 21 % |
| MOLBACE | $6 \pm 1$ s | 42 % |
| MOLLIPO | $14 \pm 1$ s | 38 % |
| MOLTOX21 | $15 \pm 1$ s | 27 % |
| MOLTOXCAST | $16 \pm 1$ s | 13 % |
| ZINC | $44 \pm 1$ s | 27 % |
| MOLHIV | $152 \pm 1$ s | 31 % |

Following the Hamiltonian cycle in one direction and concatenating the HALs gives a sequence $S$ (and a reverse sequence $R$, for the other direction).

We say a sequence $S$ of length $n$ is a *cyclic shift* of another sequence $S'$ of length $n$ if there exists an $\ell \in \mathbb{N}$ such that $S_i = S'_j$ for all $i = 1, \ldots, n$ where $j = i + \ell \mod n$. The HAL sequence uniquely identifies biconnected outerplanar graphs (if both directions and cyclic shifts are considered):

**Lemma 1** (Colbourn & Booth (1981)). *Two biconnected outerplanar graphs $G$ and $H$ with HAL and reverse sequences $S_G$, $S_H$ and $R_G$, $R_H$ are isomorphic, iff $S_G$ is a cyclic shift of $S_H$ or $R_H$.*

## 3 IDENTIFYING OUTERPLANAR GRAPHS USING WEISFEILER-LEMAN

We develop a graph transformation called *cyclic adjacency transform* (CAT), that enables WL to distinguish all outerplanar graphs. We first introduce $\text{CAT}^*$, enabling WL to distinguish any biconnected outerplanar graphs, and then extend it to all outerplanar graphs.

In $\text{CAT}^*$ (see Section 3.1), nodes are duplicated to represent the Hamiltonian cycle in both directions. We annotate edges outside of the Hamiltonian cycle with the distance of the endpoints of the edge. This allows the Weisfeiler-Leman algorithm to encode the HAL sequence in the unfolding trees of the nodes and in turn distinguish non-isomorphic biconnected outerplanar graphs.

For extending our transformation to all outerplanar graphs (in Section 3.2), we need to ensure, that the biconnected components keep their unique encoding and also that the attachment point is encoded uniquely for isomorphic biconnected components (biconnected components might be rotated, leading to non-isomorphic graphs, that have the same components). We do that by introducing articulation and block pooling vertices. The whole graph can then be encoded as a tree, on which the Weisfeiler-Leman algorithm is maximally expressive.

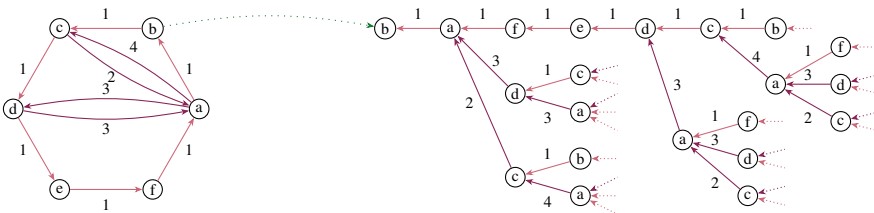

Figure 3: One part of the CAT* transformation of the graph from Figure 2 and an example unfolding tree of one of its nodes from which the HAL sequence of the original graph can be reconstructed.

### 3.1 IDENTIFYING BICONNECTED OUTERPLANAR GRAPHS USING WEISFEILER-LEMAN

We first present a graph transformation called CAT*, that allows the Weisfeiler-Leman algorithm to distinguish any two *biconnected* outerplanar graphs. Figure 2 shows an example of CAT*. Note that CAT*(G) consists of two disjoint copies of G, with directed and annotated edges.

**Definition 1.** *The CAT\* transformation takes a biconnected outerplanar graph $G = (V, E, \mu, \nu)$ and yields a modified graph $G' = \text{CAT}^*(G) = (V', E', \mu', \nu')$ by performing the steps below.*

1. *Let $C = (v_1, \ldots, v_n)$ be a (directed) Hamiltonian cycle of $G$ and $\overleftarrow{C}$ be its reverse.*

2. *Add node disjoint copies of $C$ and $\overleftarrow{C}$ to $G'$ and set $\nu'(e) = (1, \nu(e))$ for all edges in $G'$.*

3. *Let $D \subseteq E$ be the edges of $G$ not on the (undirected) Hamiltonian cycle. Add edges in both directions to $G'$ for the copies of $C$ and $\overleftarrow{C}$ for each edge in $D$: $E(G') = E(G') \cup E_d \cup E_{\overleftarrow{d}}$ with $E_d = \bigcup_{\{v_i, v_j\} \in D} \{(v'_i, v'_j), (v'_j, v'_i)\}$ and $E_{\overleftarrow{d}} = \bigcup_{\{v_i, v_j\} \in D} \{(v''_i, v''_j), (v''_j, v''_i)\}$ for copies $v'_i$ of $v_i$ in $C$ (resp. $v''_i$ in $\overleftarrow{C}$). Set $\mu'(v'_i) = \mu(v_i)$ and $\mu'(v''_i) = \mu(v_i)$ for the nodes in $G'$.*

4. *$\forall (v_i, v_j) \in E_d$ set $\nu'(v_i, v_j) = (d_C(v_j, v_i), \nu(v_i, v_j))$ and $\forall (v'_i, v'_j) \in E_{\overleftarrow{d}}$ set $\nu'(v'_i, v'_j) = (d_{\overleftarrow{C}}(v'_j, v'_i), \nu(v_j, v_i))$.*

Using CAT* we prove our first main result.

**Theorem 1.** *Two biconnected outerplanar graphs $G$ and $H$ are isomorphic, if and only if $\text{WL}(\text{CAT}^*(G)) = \text{WL}(\text{CAT}^*(H))$.*

*Proof.* Two graphs are distinguished by WL iff the multisets of node colors of their stable colorings differ. Trivially, $|V(G)| \neq |V(H)| \Rightarrow |V(\text{CAT}^*(G))| \neq |V(\text{CAT}^*(H))| \Rightarrow \text{WL}(\text{CAT}^*(G)) \neq \text{WL}(\text{CAT}^*(H))$, so we only focus on graphs with $|V(G)| = |V(H)|$. Two nodes only get the same color, if their unfolding trees are isomorphic. The first number in the HAL of each node is always 1, so it can be ignored, and the last number is always $|V(G)| - 1$, so this can simply be reconstructed by $|V(\text{CAT}^*(G))|$. The rest of the HAL sequence and the node labels of $G$ can be reconstructed from the unfolding tree of any node in $\text{CAT}^*(G)$: Trivially, each node has two direct neighbors in the Hamiltonian cycle. In the unfolding tree these are the parent and the single child with the 1-annotated edge. All other neighbors in the HAL can be reconstructed by looking at the weights of the edges that do not have weight 1. Figure 3 shows an example. Looking at any two biconnected outerplanar graphs with $n$ nodes, Weisfeiler-Leman will be able to distinguish them after at most $n$ iterations, iff they are non-isomorphic: Since the HAL sequence is encoded in the unfolding trees from all starting points (cyclic shift) and, because of the reverse copy, in both directions (reverse direction), this identifies isomorphism by Lemma 1. □

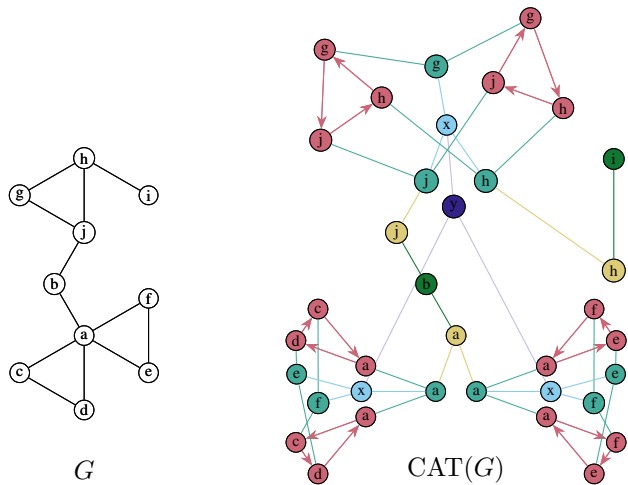

Figure 4: A graph and its CAT transformation. Original node labels are represented by letters, edge and node labels from the CAT transform are represented by colors. $\mathrm{CAT}(G)$ looks like a cat.

## 3.2 THE CAT TRANSFORMATION

We define the CAT transformation by applying $\mathrm{CAT}^*$ to the blocks of the graphs and adding nodes and edges to make outerplanar graphs distinguishable by WL.

**Definition 2.** *The* $\mathrm{CAT}(G) = G'$ *transformation maps a graph* $G$ *to a new graph* $G'$ *as follows:*

1. *Let* $B_1, \ldots, B_\ell$ *be the blocks of* $G$ *and let* $F$ *be the graph induced by the edges of* $G$ *that are not in any block. Let* $\{\bot, \square, \bowtie, \star, \triangle\}$ *be distinct node labels not in* $X$.

2. *Add* $F$ *to* $G'$ *with labels* $\mu'(v) = (\bot, \mu(v))$ *for* $v \in F$.

3. *For each block* $B_i$ *in* $G$:

    3.1. *Let* $B_i', \overleftarrow{B_i'}$ *be the two connected components in* $\mathrm{CAT}^*(B_i)$. *Add* $B_i'$ *and* $\overleftarrow{B_i'}$ *to* $G'$.

    3.2. *Let* $A_i = V(B_i) \cap V(F)$ *be the nodes of* $B_i$ *in* $F$.

    3.3. *Let* $\gamma_i : A_i \to V(B_i')$ *map nodes of* $F$ *to their copy in* $B_i'$ *and* $\overleftarrow{\gamma_i}$ *to the copy in* $\overleftarrow{B_i'}$.

    3.4. *For all pairs* $(v, \overleftarrow{v})$ *of corresponding nodes in* $B_i'$ *and* $\overleftarrow{B_i'}$ *add a node* $p_v$ *with* $\mu'(p_v) = (\star, \mu(v))$ *and edges* $\{p_v, v\}, \{p_v, \overleftarrow{v}\}$ *to* $G'$.

    3.5. *Add a node* $b_i$ *to* $G'$ *with* $\mu'(b_i) = \square$. *For all* $v \in V(B_i)$ *add an edge* $\{b_i, p_v\}$.

    3.6. *For each* $a \in A_i$ *let* $\mu'(a) = (\bowtie, \mu(a))$ *and add edge* $\{p_{\gamma_i(a)}, a\}$ *to* $G'$.

4. *Add a node* $g$ *with* $\mu'(g) = \triangle$ *to* $G'$ *and for all nodes* $b_i$, *add an edge* $\{g, b_i\}$ *to* $G'$.

5. *Let* $\mathrm{CAT}(G) = G'$.

An example of the CAT transformation can be seen in Figure 4. Appendix C contains additional visualizations of the transformation on real-life molecular graphs. We refer to nodes created in step 3.1 as Hamiltonian cycle nodes, those created in step 3.4 as pooling nodes, those created in step 3.5 as block nodes and those created in step 3.6 as articulation nodes. Finally, the node created in step 4 is called the global (block) pooling node.

**Theorem 2.** *Outerplanar graphs* $G$ *and* $H$ *are isomorphic, iff* $\mathrm{WL}(\mathrm{CAT}(G)) = \mathrm{WL}(\mathrm{CAT}(H))$.

*Proof.* Following Theorem 1, each block will be uniquely identified by WL. Since the additional nodes have distinct labels, they will not cause WL to falsely report two blocks as isomorphic when they are not. The information about the entire HAL sequence of each block is stored in the $b$ nodes

Table 3: Resistance and diameter before and after the CAT transformation. $\overline{\rho(G)}$ and $\max_{\rho(G)}$ denote the average and maximum pair-wise effective resistance for graph $G$. Results are reported as mean and standard deviation across all graphs in the datasets. In all cases, smaller is better.

| Dataset | $\Phi(G)$ | $\Phi(\text{CAT}(G))$ | $\overline{\rho(G)}$ | $\overline{\rho(\text{CAT}(G))}$ | $\max_{\rho(G)}$ | $\max_{\rho(\text{CAT}(G))}$ |
|---|---|---|---|---|---|---|
| ZINC | $12.5 \pm 2.6$ | $\mathbf{9.9 \pm 1.6}$ | $4.0 \pm 0.7$ | $\mathbf{2.6 \pm 0.4}$ | $10.0 \pm 2.0$ | $\mathbf{7.7 \pm 1.9}$ |
| MOLESOL | $\mathbf{6.6 \pm 3.3}$ | $6.9 \pm 3.8$ | $2.3 \pm 1.0$ | $\mathbf{2.1 \pm 0.9}$ | $5.5 \pm 2.3$ | $6.0 \pm 2.8$ |
| MOLTOXCAST | $8.5 \pm 4.7$ | $\mathbf{8.4 \pm 4.0}$ | $3.0 \pm 1.5$ | $\mathbf{2.6 \pm 1.3}$ | $7.2 \pm 4.3$ | $7.4 \pm 3.9$ |
| MOLTOX21 | $8.8 \pm 4.6$ | $\mathbf{8.7 \pm 4.0}$ | $3.1 \pm 1.5$ | $\mathbf{2.7 \pm 1.3}$ | $7.5 \pm 4.2$ | $7.7 \pm 3.9$ |
| MOLLIPO | $13.8 \pm 4.0$ | $\mathbf{9.9 \pm 2.1}$ | $4.3 \pm 1.2$ | $\mathbf{2.6 \pm 0.5}$ | $10.7 \pm 3.4$ | $\mathbf{7.9 \pm 2.3}$ |
| MOLBACE | $15.1 \pm 3.2$ | $\mathbf{11.5 \pm 2.8}$ | $5.0 \pm 1.3$ | $\mathbf{2.9 \pm 0.7}$ | $12.5 \pm 3.4$ | $\mathbf{9.1 \pm 2.6}$ |
| MOLSIDER | $12.6 \pm 11.8$ | $\mathbf{10.4 \pm 7.3}$ | $4.1 \pm 3.8$ | $\mathbf{2.9 \pm 2.2}$ | $10.4 \pm 11.0$ | $\mathbf{8.9 \pm 6.8}$ |
| MOLBBBP | $10.7 \pm 3.7$ | $\mathbf{9.1 \pm 2.6}$ | $3.4 \pm 1.1$ | $\mathbf{2.4 \pm 0.6}$ | $8.3 \pm 3.8$ | $\mathbf{7.5 \pm 2.5}$ |
| MOLHIV | $11.9 \pm 5.2$ | $\mathbf{9.9 \pm 3.8}$ | $3.9 \pm 1.7$ | $\mathbf{2.7 \pm 1.2}$ | $9.3 \pm 4.7$ | $\mathbf{8.2 \pm 3.8}$ |

after some iteration. The $p$ nodes connect the block and $b$ nodes to the rest of the graph (through the $a$ nodes), determining the orientation of the block. Note that the graph returned by CAT without the CAT$^*$ blocks and the node $g$ is a tree. Relying on the labels of the $p$ and $b$ nodes, we can reconstruct the original graph from this tree. As WL can distinguish labeled trees (Arvind et al., 2015; Kiefer, 2020), it can thus distinguish non-isomorphic outerplanar graphs using CAT. For the other direction, note that CAT is permutation-invariant: for two isomorphic graphs $G$ and $H$, the graphs $\text{CAT}(G)$ and $\text{CAT}(H)$ are isomorphic and WL will give the same output for both. □

Importantly, we can compute $\text{CAT}(G)$ in linear time. The computational complexity is dominated by the computation of the blocks (Tarjan, 1972) and their Hamiltonian cycles (Mitchell, 1979), which both require linear time. Note that we only add a linear number of nodes and edges. From Morris et al. (2019) and Xu et al. (2019) it follows, that MPNNs that are as expressive as 1-WL can distinguish $\text{CAT}(G)$ and $\text{CAT}(H)$ for non-isomorphic outerplanar graphs $G$ and $H$. Thus, we propose to transform the input graphs using CAT and apply an MPNN on them.

Note that our proof used an important property of the WL algorithm: Adding nodes and edges to WL-distinguishable graphs does never lead to WL-indistinguishable graphs, as long as new labels are used. We use this to add a global pooling node in step 4 of CAT which is connected to all block pooling nodes. This allows to pass messages between block nodes in fewer iterations in the subsequent MPNN step. In the next section we analyze the effect of CAT on the connectivity of the transformed graph.

CAT can also be applied to non-outerplanar graphs. In this case, our graph transformation performs the steps described in Definition 2. However, if a non-outerplanar block $B_i$ is encountered, only one copy $B_i'$ is created in $\text{CAT}(G)$ and its vertices are connected to the corresponding pooling nodes. While this never reduces expressivity, it is also not guaranteed to improve expressivity on non-outerplanar graphs. Note that it can be determined in linear time whether a block is outerplanar, while trying to compute the Hamiltonian cycle of the block (Mitchell, 1979). Hence, the CAT transformation always only requires linear time.

### 3.3 INFLUENCE OF CAT ON GRAPH CONNECTIVITY

We investigate the effects of CAT on different measures of graph connectivity. GNNs can in fact be susceptible to poor performance, e.g., in tasks that depend on long-range interactions (Alon & Yahav, 2021). We are therefore interested in analyzing whether CAT improves graph connectivity measures such as the diameter $\Phi(G)$ of a graph $G$. Here, we refer to the shortest path distance $d$ between two nodes as the *distance* between them. We use $d(a, b)$ to denote the distance between two nodes $a, b$ in $\text{CAT}(G)$ and $d_G(a, b)$ to denote distance between two nodes $a, b$ in $G$.

**Observation 1.** *Let $B$ be a block of a graph $G$, it holds that $\Phi(\text{CAT}(B)) \leq 4$.*

*Proof.* Let $a, b \in V(\text{CAT}(B))$. By definition all nodes in $\text{CAT}(B)$ are either from a Hamiltonian cycle created by CAT$^*$, a pooling node or a block node. If both nodes are from a Hamiltonian cycle, then there is a path $a, p_a, b_B, p_b, b$ between them, where $p_a$ / $p_b$ are pooling nodes and $b_B$ is the

block node. Hence, $d(a, b) \leq 4$. If $a$ or $b$ is a pooling or a block node, then the above path implies that $d(a, b) < 4$. □

**Observation 2.** *Let $B_i$ and $B_j$ be two blocks of a graph $G$. In $\mathrm{CAT}(G)$, the maximum distance between any node in $\mathrm{CAT}(B_i)$ and any node in $\mathrm{CAT}(B_j)$ is 6.*

*Proof.* Let $a \in V(\mathrm{CAT}(B_i))$ and $b \in V(\mathrm{CAT}(B_j))$. If $B_i = B_j$, then Observation 1 implies $d(a, b) \leq 4$. If $B_i \neq B_j$, then there exists a path $a, p_a, b_i, g, b_j, p_b, b$ where $p_a$ / $p_b$ is a pooling node for $a$ / $b$, $b_i$ / $b_j$ is the block node for block $B_i$ / $B_j$, and $g$ is the global block pooling node. Thus, $d(a, b) \leq 6$. □

**Proposition 1.** *For an outerplanar graph $G$, $\Phi(\mathrm{CAT}(G)) \leq \Phi(G) + 7$.*

*Proof sketch.* To prove the proposition we analyze the distance between any pairs of nodes in $\mathrm{CAT}(G)$ by case analysis on the type of nodes and show that any two nodes in $\mathrm{CAT}(G)$ can be at most at distance $\Phi(G) + 7$. We defer the full proof to Appendix A. □

Proposition 1 states that for outerplanar graphs, in the worst case, CAT increases the graph diameter by at most a constant additive factor of 7. We claim that in most practical cases the short-cutting inside or between blocks should lead to CAT consistently reducing the graph diameter (see Observations 1 and 2). In Table 3, we demonstrate this on molecular benchmark datasets. Besides the diameter, another useful graph connectivity measure is the *effective resistance*. The notion of effective resistance originates in electrical engineering (Kirchhoff, 1847) and has implications on several graph properties. For example, the effective resistance between two nodes is proportional to the commute time between them (Chandra et al., 1989). Intuitively, a large effective resistance between two nodes suggests that information propagation between the nodes is hindered. Recently, effective resistance has been in fact linked to *over-squashing* (Black et al., 2023) in GNNs, which is a negative effect that leads to long-range interactions having little impact on the predictions of a GNN. Effective resistance as introduced by Kirchhoff (1847) is naturally only defined for undirected graphs. As CAT produces directed graphs, we therefore use an extension of effective resistance introduced by Young et al. (2015) that is applicable to directed graphs. We refer to Young et al. (2015) for more details. In Table 3 we demonstrate that CAT reduces the pair-wise effective resistance on molecular benchmark datasets.

## 4    DISCUSSION AND RELATED WORK

It is well known that the expressivity of MPNNs is bounded by the 1-WL test (Morris et al., 2019; Xu et al., 2019). This means that any pair of non isomorphic graphs that cannot be distinguished by 1-WL will get mapped to the same embedding by any MPNN. One such pair of graphs are decalin and bicyclopentyl molecules (see Fig. 4). As these two graphs are outerplanar it follows that MPNNs are not sufficiently expressive for outerplanar graphs. Furthermore, in the graph mining community it is well known that many pharmaceutical molecules are outerplanar (Horváth et al., 2006; Horváth & Ramon, 2010). Outerplanarity has also been discussed in the context of reconstruction with GNNs (Cotta et al., 2021). This motivates the need for GNNs that are highly expressive on outerplanar graphs. Outerplanar graphs have treewidth at most two (Bodlaender, 1998) and Kiefer (2020) showed that 3-WL is sufficiently expressive to distinguish all outerplanar graphs. Hence, any GNN which matches the expressivity of 3-WL, such as 3-IGN (Maron et al., 2019) or 3-GNN (Morris et al., 2019), is capable of solving

Bicyclopentyl

Decalin

Figure 5: Two molecules corresponding to outerplanar graphs that cannot be distinguished by 1-WL.

our main goal of distinguishing all outerplanar graphs. However, the runtime of the 3-WL test is $\mathcal{O}(n^3 \log n)$, which can be infeasible for even medium-sized real-world graphs (Immerman & Lander, 1990; Kiefer, 2020). Similarly, 3-GNN and 3-IGN run in roughly $\mathcal{O}(n^3)$ time (Maron et al., 2019; Morris et al., 2019). Thus, there currently exists no practical GNN architecture which can distinguish all outerplanar graphs. Even when additionally restricting the graph class to *biconnected*

Table 4: Predictive performance of MPNNs with and without CAT on different molecular benchmark datasets. Arrows indicate whether smaller ($\downarrow$) or bigger ($\uparrow$) results are better. **Bold** entries are an MPNN with CAT that outperforms the same MPNN without CAT.

| Dataset $\rightarrow$ $\downarrow$ Model | ZINC MAE $\downarrow$ | MOLHIV ROC-AUC $\uparrow$ | MOLBACE ROC-AUC $\uparrow$ | MOLBBBP ROC-AUC $\uparrow$ | MOLSIDER ROC-AUC $\uparrow$ |
|---|---|---|---|---|---|
| GIN | $0.168 \pm 0.007$ | $77.9 \pm 1.0$ | $74.6 \pm 3.2$ | $66.0 \pm 2.1$ | $56.6 \pm 1.0$ |
| CAT+GIN | $\mathbf{0.101 \pm 0.004}$ | $76.7 \pm 1.8$ | $\mathbf{79.5 \pm 2.5}$ | $\mathbf{67.2 \pm 1.8}$ | $\mathbf{58.2 \pm 0.9}$ |
| GCN | $0.184 \pm 0.013$ | $76.7 \pm 1.4$ | $77.9 \pm 1.7$ | $66.1 \pm 2.4$ | $56.7 \pm 1.5$ |
| CAT+GCN | $\mathbf{0.123 \pm 0.008}$ | $\mathbf{77.1 \pm 1.6}$ | $\mathbf{79.2 \pm 1.5}$ | $\mathbf{68.3 \pm 1.7}$ | $\mathbf{57.9 \pm 1.8}$ |
| GAT | $0.375 \pm 0.013$ | $76.6 \pm 2.0$ | $81.7 \pm 2.3$ | $66.2 \pm 1.4$ | $58.4 \pm 1.0$ |
| CAT+GAT | $\mathbf{0.201 \pm 0.022}$ | $75.3 \pm 1.6$ | $79.3 \pm 1.6$ | $66.0 \pm 1.9$ | $58.3 \pm 1.3$ |

| Dataset $\rightarrow$ $\downarrow$ Model | MOLESOL RMSE $\downarrow$ | MOLTOXCAST ROC-AUC $\uparrow$ | MOLLIPO RMSE $\downarrow$ | MOLTOX21 ROC-AUC $\uparrow$ |
|---|---|---|---|---|
| GIN | $1.105 \pm 0.077$ | $65.3 \pm 0.6$ | $0.717 \pm 0.016$ | $75.8 \pm 0.7$ |
| CAT+GIN | $\mathbf{0.985 \pm 0.055}$ | $\mathbf{65.6 \pm 0.5}$ | $0.798 \pm 0.031$ | $74.8 \pm 1.0$ |
| GCN | $1.053 \pm 0.087$ | $64.4 \pm 0.4$ | $0.748 \pm 0.018$ | $76.4 \pm 0.3$ |
| CAT+GCN | $\mathbf{1.006 \pm 0.036}$ | $\mathbf{66.2 \pm 0.8}$ | $0.771 \pm 0.023$ | $74.9 \pm 0.8$ |
| GAT | $1.037 \pm 0.063$ | $63.8 \pm 0.8$ | $0.728 \pm 0.024$ | $76.3 \pm 0.6$ |
| CAT+GAT | $1.09 \pm 0.048$ | $\mathbf{64.5 \pm 0.8}$ | $0.754 \pm 0.021$ | $75.4 \pm 0.7$ |

outerplanar graphs, MPNNs are not sufficiently expressive (see Fig. 1). Furthermore, Zhang et al. (2023b) has shown that most GNNs cannot even detect simple properties associated with biconnectivity such as articulation vertices. They find that only their distance-based GNN and specific GNNs based on subgraphs (Bevilacqua et al., 2021; Frasca et al., 2022) and are able to detect some of these properties. Again, these approaches have an at least quadratic worst case runtime.

Interestingly, it often seems to be impossible to directly use outerplanarity to speed up the pre-processing of many higher-order GNNs. For example, finding a subgraph remains NP-hard for outerplanar graphs (Sysło, 1982). Thus, methods like the graph structural network (Bouritsas et al., 2022) that rely on counting subgraphs remain computationally expensive even on outerplanar graphs. Subgraph GNNs model graphs as collection of subgraphs (Frasca et al., 2022), this usually requires a pre-processing with at least quadratic runtime, depending on the method used to extract subgraphs. For example, node-delete (Bevilacqua et al., 2021) creates all subgraphs which are obtained by deleting a single node which always creates $\mathcal{O}(V^2)$ nodes and $k$-ego-net (Bevilacqua et al., 2021) extracts the $k$-hop neighborhood for each node which for $k \geq 2$ can create $\mathcal{O}(V^2)$ nodes in the worst case, for example for star graphs, which are also outerplanar.

Finally, there exist many GNNs which are provably more expressive than WL. Many proofs that show that a proposed architecture is more expressive than WL do this by arguing that the architecture is never less expressive than WL and providing a single example where it is more expressive, see for example Bodnar et al. (2021b;a), Wijesinghe & Wang (2022), and Bevilacqua et al. (2021). However, not much is known about the family of graphs which such architectures can distinguish. Also proving an upper bound on the expressivity of an architecture is considered difficult and requires significant effort as demonstrated by Zhang et al. (2023a). In contrast, we identify outerplanar graphs as a large practical graph family that our proposed method CAT can distinguish.

## 5 EXPERIMENTAL EVALUATION

We investigate whether our proposed method CAT[1] can improve the predictive performance of MPNNs on molecular benchmark datasets. We utilize three commonly used MPNNs: GIN (Xu et al., 2019), GCN (Kipf & Welling, 2017), and GAT-v2 (Veličković et al., 2018; Brody et al., 2022). We train on the commonly used ZINC (Gómez-Bombarelli et al., 2018; Sterling & Irwin, 2015) and MOLHIV (Hu et al., 2020) datasets, which contain graphs representing molecules. We supplement these with 7 small datasets (see Table 4) from the OGB collection (Hu et al., 2020). In total we train with 3 MPNNs on 9 datasets with and without CAT. For each configuration, we separately tune

---

[1]Our code can be found at `https://anonymous.4open.science/r/outerplanarGNNs`.

hyperparameters on the validation set and train a model with the best hyperparameters 10 times on the training set and evaluate it on the test set. For each dataset we report the mean and standard deviation of the most common evaluation metric on the test set in the epoch with the best validation performance. For ZINC we use a batch size of 128 and an initial learning rate of $10^{-3}$ that gets halved if the validation metric does not improve for 20 epochs. The training stops after 500 epochs or if the learning rate dips below $10^{-5}$. For all other datasets we train with a fixed learning rate for 100 epochs and a batch size of 128. We use the same hyperparameter grid for all models and provide more details in Appendix B. Besides measuring the predictive performance, we also measure the time needed for applying CAT (averaged over 10 runs), and the training and evaluation time for GIN and GIN+CAT with the same hyperparameters on all datasets (averaged over 5 runs). Finally, we report the values for the diameters and effective resistances as described in Section 3.3.

**Results.** Table 2 shows the pre-processing time of CAT. Note that this is the performance of running CAT on only a single CPU core. Thus, it is possible to achieve faster runtimes by simply parallelizing different graphs over different cores. This negligible runtime of around 5ms per graph on MOLHIV allows to apply the transformation even in realistic high-throughput screening applications (Krasoulis et al., 2022). Training and prediction time on CAT transformed graphs increases by 29% on average. Table 4 shows the predictive performance of all models. Note that our baseline models obtain very strong results, often surpassing the performance of (higher-order) GNNs reported in the literature and that we train each MPNN and MPNN+CAT with exactly the same sets of hyperparameters. Overall, CAT improves the predictive performance of GIN and GCN in the majority of datasets (6 / 9 and 7 / 9, respectively). For GIN and GCN, performance increases reliably on all datasets, except MOLLIPO and MOLTOX21. Surprisingly, CAT does not work well with GAT and only improves its performance in 2 / 9 datasets. Most notably on ZINC, CAT achieves very strong results boosting the predictive performance of MPNNs between 33% (GCN) and 46% (GAT). This is only surpassed by higher-order GNNs such as CW Networks (Bodnar et al., 2021a) which obtains a MAE of $0.079 \pm 0.006$ at the cost of potentially exponential pre-processing runtime due to enumerating cycles in the graph. Table 3 shows that CAT reduces both graph diameter and maximum pair-wise resistance on most datasets. Furthermore, CAT reduces the average pair-wise resistance in all datasets. This suggests that CAT is effective at improving graph connectivity in real-life molecular graphs.

## 6 CONCLUSION

We proposed CAT, a graph transformation that enables the Weisfeiler-Leman algorithm to be maximally expressive on outerplanar graphs. We rely on the fact that biconnected outerplanar graphs can be uniquely identified by their Hamiltonian adjacency list sequences, which CAT encodes in unfolding trees. By combining MPNNs with CAT we enable them to distinguish all outerplanar graphs as well. We achieved promising empirical results on standard molecular benchmark datasets where CAT typically improved the performance of GIN and GCN, while for GAT we could not observe this benefit. Computing CAT takes linear time and our implementation of CAT typically runs in the order of seconds, even for MOLHIV the total single-thread runtime is only 2.5 minutes. We also studied the effect of CAT on graph connectivity motivated by the recent interest in the over-squashing phenomenon. We theoretically prove that in the worst-case CAT increases the diameter of outerplanar graphs by a small additive constant. However, inspecting CAT on real-world data, we find that the diameter decreases most of the time. Similarly, we observed that the maximum and average pair-wise effective resistance, which is associated with over-squashing, typically decreases after applying CAT.

As future work we consider to further extend our approach to more general graphs families. A promising candidate are $k$-outerplanar graphs, which are known to capture even more molecular graphs (Horváth et al., 2010). A major challenge going beyond outerplanar graphs is that non-outerplanar biconnected components can have multiple or even no Hamiltonian cycles, making an extension of CAT to such graphs non-trivial. A possible approach could be to split a graph into trees and components with a unique Hamiltonian cycles. We hypothesize that such a transformation would lead to maximally expressive MPNNs at the cost of potentially exponential pre-processing time depending on the graph type.

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

## A    PROOF OF PROPOSITION 1

We prove Proposition 1 which states that $\Phi(\mathrm{CAT}(G)) \geq \Phi(G) + 7$ for every outerplanar graph $G$.

*Proof.* Let $a, b \in V(\mathrm{CAT}(G))$ such that $d(a, b) = \Phi(\mathrm{CAT}(G))$. We call a node a *tree node* if it was not part of a block in $G$ and was not created by CAT. A node that is *not* a tree node is either a Hamiltonian cycle node, a pooling node, a block pooling node, or a global block pooling node.

**Case 1:** Both $a, b$ are not tree nodes. By Observation 2: $d(a, b) \leq 6$.

**Case 2:** Node $a$ is a tree node and $b$ is not. Let $x \in V(G)$ be the closest articulation node to $a$ in $\mathrm{CAT}(G)$. Then, there is a path of length $d_G(a, x)$ in $\mathrm{CAT}(G)$ from $a$ to $x$. We can extend this path by one node to reach a pooling node. By Definition 2, there exists a path of length at most 6 from this pooling node to $b$. Thus $d(a, b) \leq d_G(a, x) + 7 \leq \Phi(G) + 7$.

**Case 3:** Both $a, b$ are tree nodes.

**Case 3a:** Suppose that the shortest path between $a$ and $b$ in $G$ does not contain any edge inside of an outerplanar block, then $d(a, b) = d_G(a, b) \leq \Phi(G)$.

**Case 3b:** Suppose that the shortest path between $a$ and $b$ in $G$ contains one or more edges inside exactly one block. Then, we can enter and exit this block in $\mathrm{CAT}(G)$ through a path $r_1, p_1, b, p_2, r_2$, where $r_1, r_2$ are articulation nodes, $p_1, p_2$ are pooling nodes, and $b$ is a block node. Note that the articulation nodes were part of the path in $G$ which implies $d(a, b) = d_G(a, r_1) + d_G(r_2, b) + 4 = d_G(a, b) - d_G(r_1, r_2) + 4$. Furthermore, we do not need to take the one or more edges inside the block to go from $r_1$ to $r_2$. Using $d_G(r_1, r_2) \geq 1$ we obtain $d(a, b) = d_G(a, b) - d_G(r_1, r_2) + 4 \leq d_G(a, b) + 3 \leq \Phi(G) + 3$.

**Case 3c:** Suppose that the shortest path between $a$ and $b$ in $G$ contains two or more edges that are contained in two or more different blocks. Then, for $\mathrm{CAT}(G)$ we can shortcut from the first to the last block node of the shortest path between $a$ and $b$ through a path $r_1, p_1, b_1, g, b_2, p_2, r_2$, where $r_1, r_2$ are articulation nodes, $p_1, p_2$ are pooling nodes, $b_1, b_2$ are block nodes, and $g$ is the global block pooling node. Note that the articulation nodes were part of the path in $G$ which implies that $d(a, b) = d_G(a, r_1) + d_G(r_2, b) + 6 = d_G(a, b) - d_G(r_1, r_2) + 6$. By assumption, we know that $d_G(r_1, r_2) \geq 2$ which implies $d(a, b) = d_G(a, b) - d_G(r_1, r_2) + 6 \leq d_G(a, b) + 4 \leq \Phi(G) + 4$.  $\square$

## B    DETAILS FOR EXPERIMENTAL EVALUATION

Our models are implemented in PyTorch-Geometric (Fey & Lenssen, 2019) and trained on a single NVIDIA GeForce RTX 3080 GPU. We use WandB (Biewald, 2020) for tracking. The used server has 64 GB of RAM, has an 11th Gen Intel(R) Core(TM) i9-11900KF CPU running at 3.50GHz. Table 5 shows the hyperparameters for our MPNNs on different datasets. We use the same hyperparameter grid for MPNNs combined with CAT. We used a smaller hyperparameter grid for MOLHIV than for ZINC, as MOLHIV is larger than ZINC meaning that training takes much longer. When benchmarking the speed of GIN against GIN+CAT we train for 100 epochs with a batch size of 128 on all datasets with the same hyperparameters for both models (see Table 5).

CAT **implementation.**    CAT adds an additional feature to each node which encodes the type of that node i.e., nodes from Hamiltonian cycles, block nodes, pooling nodes, articulation nodes and or global block nodes. Furthermore, we create additional edge features encoding the types of nodes incident to this edge i.e., an edge between two different nodes in a Hamiltonian cycle has a different type than an edge from a pooling node to the block node. For newly created nodes and edges we set their remaining features to the feature of the node / edge they are based on; for example, a pooling node will have the features of the node they are performing the pooling operation for. For nodes that have no natural representation in the graph (block and block pooling nodes) we set these features to 0. To ensure that only these nodes get assigned 0 features, we shift the values of these features for all other nodes by 1. Note that our MPNNs treat the distance on edges in blocks as a categorical feature. Representing the distances as numerical features did not improve performance in preliminary experiments.

Table 5: Hyperparameter grids for GIN, GCN and GAT on different datasets.

| Parameter | All datasets except `MOLHIV` | `MOLHIV` | Benchmarking GIN (+CAT) all datasets |
|---|---|---|---|
| Message passing layers | 2, 3, 4, 5 | 4, 5 | 4 |
| Final MLP layers | 2 | 2 | 2 |
| Pooling operation | mean, sum | mean, sum | mean |
| Embedding dimension | 64, 128, 256 | 64,128 | 64 |
| Jumping knowledge | last | concat | concat |
| Dropout rate | 0, 0.5 | 0.5 | 0 |

## C    ADDITIONAL FIGURES

We provide additional visualizations of the CAT transformation. In all figures, the color of the vertices in the transformed graph have the following meaning: red nodes are from Hamiltonian cycles, blue nodes correspond to blocks, yellow nodes pool the nodes from Hamiltonian cycles, orange nodes correspond to articulation nodes and the gray node pools block nodes. Figure 6 shows a synthetic example with a non-outerplanar graph. Figure 7 demonstrates CAT on various synthetic graphs. Figure 8 shows the result of CAT on molecular graphs from `ZINC` and `MOLHIV`. Somewhat ironically, CAT often generates frog graphs on `MOLHIV` as can be seen in Figure 9.

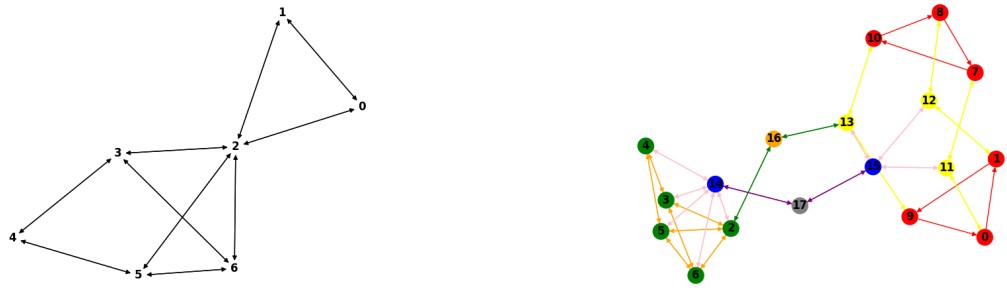

Figure 6: Left: example non-outerplanar graph. Right: result of applying CAT to the graph. Colors indicate the type of node (see Appendix C).

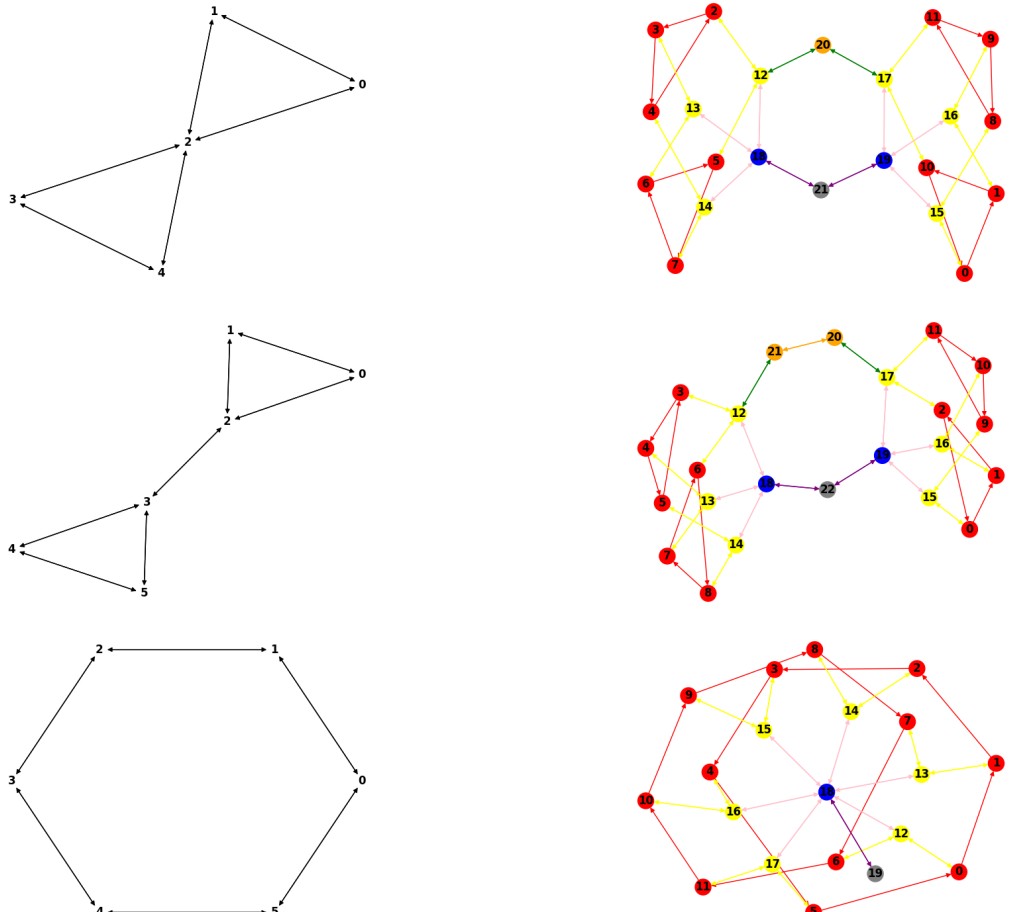

Figure 7: Left: example graphs. Right: result of applying CAT to these graphs. Colors indicate the type of node (see Appendix C).

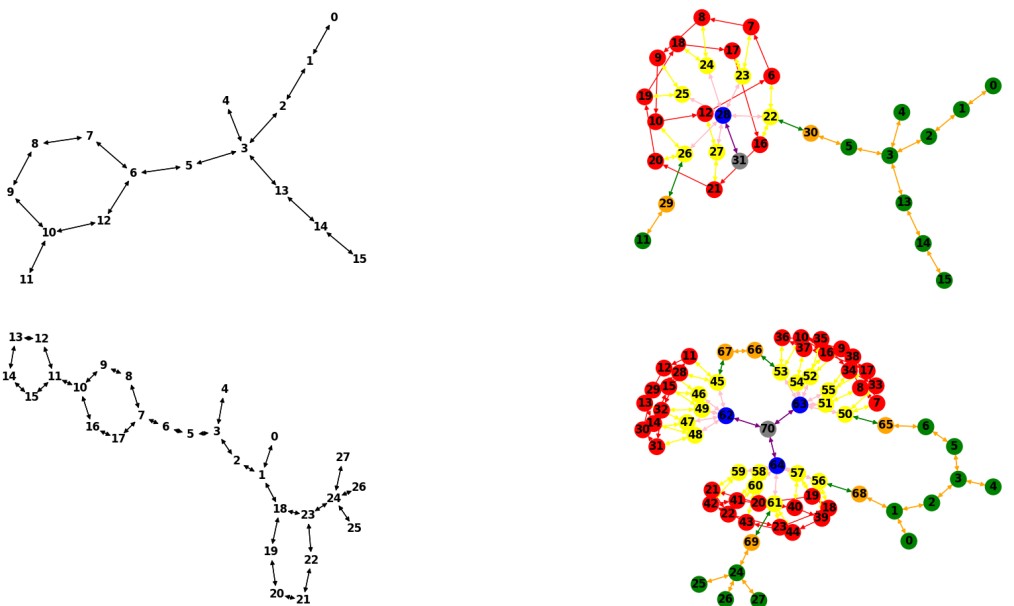

Figure 8: Left: example graphs from MOLHIV (top) and ZINC (bottom). Right: result of applying CAT to these graphs. Colors indicate the type of node (see Appendix C).

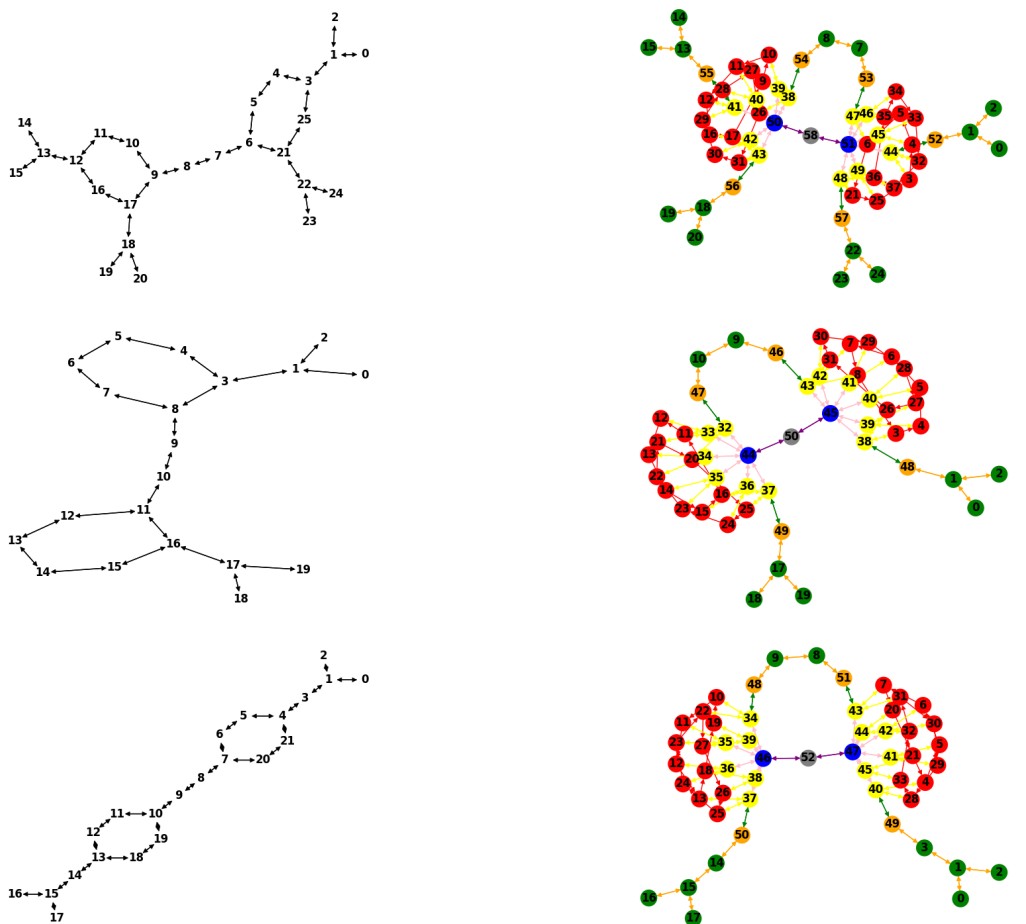

Figure 9: Left: example graphs from MOLHIV. Right: result of applying CAT to the graph. Colors indicate the type of node (see Appendix C).

