# OpenReview forum: "Maximally Expressive GNNs for Outerplanar Graphs"
_ICLR.cc/2024/Conference — Submitted to ICLR 2024_

### Official Review · Reviewer_vp7g · 2023-10-16

**Soundness:** 3 good
**Presentation:** 3 good
**Contribution:** 3 good
**Rating:** 6
**Confidence:** 3

**Summary:**

This paper proposed a graph transformation method CAT for outerplanar graphs that can run in linear time. Authors show that graphs after transformation can be fully distinguished by 1-WL, resulting in a simple MPNN that can be maximally expressive for it. Since most real-world molecular graphs are outerplanar graphs, the proposed method has great potential for related downstream tasks, avoiding the high computational cost introduced by traditional high-expressive GNNs. Authors compare their transformation with the original one on several real-world datasets and demonstrate the improvement.

**Strengths:**

1. the proposed transformation is sound and its practical runtime is promising.
2. Authors theoretically prove the MPNN can be maximally expressive for the CAT-transformed outerplanar graphs.
3. The proposed method can reduce the diameter and effective resistance of graphs, which could bring performance improvement for tasks that require long-range information.
4. The comparison result between the datasets before and after the transformation is promising.

**Weaknesses:**

1. Authors only compare the CAT with the non-CAT version on MPNN. I believe more baseline models need to be included for completeness. It would be great to see pre-processing cost + training/inference cost + performance comparison on different baseline methods like subgraph-based GNNs and high-order GNNs.
2. Authors claim CAT can help alleviate the problem of over-squashing and improve the performance of long-range tasks. However, no evaluations are performed. I believe some experiments should be conducted, like datasets in Long-Range Benchmarks [1].


[1] Dwivedi et al., Long Range Graph Benchmark, NeurIPS22, Track on Datasets and Benchmarks.

**Questions:**

1. Can the authors explain why the CAT achieved worse performance than the original one in MOLLIPO and MOLTOX21? Especially in MOLLIPO, the gap is significant.

---

> ### Author Response · Authors · 2023-11-15
>
> Thank you very much for your review and comments.
>
> **Weakness 1:**
>
> > Authors only compare the CAT with the non-CAT version on MPNN. I believe more baseline models need to be included for completeness. It would be great to see pre-processing cost + training/inference cost + performance comparison on different baseline methods like subgraph-based GNNs and high-order GNNs.
>
> Please refer to the global response regarding additional experiments. We intend to include at least one additional baseline and include these metrics.
>
> **Weakness 2:**
> > Authors claim CAT can help alleviate the problem of over-squashing and improve the performance of long-range tasks. However, no evaluations are performed. I believe some experiments should be conducted, like datasets in Long-Range Benchmarks [1].
>
> In the current submission, we focus on maximizing the expressivity on outerplanar graphs. In Table 3 we provide first evidence that $\text{CAT}$ typically reduces the diameter and the effective resistance of the graphs. However, as $\text{CAT}$ is not designed specifically for this, we see further investigation into whether $\text{CAT}$ can help reduce oversquashing and long range tasks in general as future work.
>
> **Question 1:**
> >Can the authors explain why the CAT achieved worse performance than the original one in MOLLIPO and MOLTOX21?
>
> Unfortunately, we do not have insights on why $\text{CAT}$ performs worse on these datasets. Overall it is difficult to say why certain models do not perform well on certain datasets and it often requires more insight in both the dataset and the task at hand, as well as the strengths and weaknesses of the model. Our goal was not to achieve state-of-the-art performance on all datasets. Instead we show that $\text{CAT}$ in most cases increases the performance for commonly used datasets with outerplanar graphs (ZINC and the $8$ MOL- datasets from OGB).

---

> > ### Comment · Reviewer_vp7g · 2023-11-20
> > **Comments on Authors' answers**
> >
> > Thanks for the answers to all my concerns. I checked comments from all reviewers and authors' replies. Currently, I cannot see enough evidence to either increase or decrease my score, and would like to maintain it.

---

> > > ### Author Response · Authors · 2023-11-20
> > >
> > > Thank you for keeping us updated!

---

### Official Review · Reviewer_geBr · 2023-10-31

**Soundness:** 2 fair
**Presentation:** 2 fair
**Contribution:** 1 poor
**Rating:** 3
**Confidence:** 5

**Summary:**

This paper proposes a graph transformation (CAT) that enhances the expressive power of standard graph neural networks (GNN) on outerplanar graphs: CAT-enhanced GNNs are capable of distinguishing all non-isomorphic outerplanar graphs. The authors rely on a specific property of outerplanar graphs: biconnected outerplanar graphs have a unique hamiltonian cycle which can be identified in linear time, and this by identifying this cycle, one can uniquely identify each biconnected outerplanar graph. The authors hence use a linear-time algorithm to identify the cycle, and then compute the hamiltonian adjacency list (HAL), which is then used to create an annotated transformation of the input graph. Since 1-WL-expressive GNNs can identify any labelled trees, the authors show that the presented algorithm combined with a 1-WL-expressive-GNN can identify outerplanar graphs. Experiments on two molecular datasets shows that the method yields meaningful improvements on the baseline GNN models.

**Strengths:**

**Natural idea**: The idea of enhancing GNNs with easy-to-compute structural context is natural, and in this instance, it has implications of a subset of planar graphs in that it leads to a complete algorithm on outerplanar graphs.

**Limited computational overhead**: The approach boils down to a simple pre-processing step, but once the pre-processing is done it can used with any GNN model.

**Weaknesses:**

- **Scholarship**: The fundamental of this paper is to annotate graphs with labels to enhance the discriminative capacity of standard GNNs, which is widely studied under different node labelling approaches and a more thorough related work coverage is essential. Most importantly, there is a recent paper "Dimitrov et al, PlanE: Representation Learning on Planar Graphs, NeurIPS 2023"  which introduces a *complete algorithm on the class of all planar graphs*. This paper strictly subsumes the result presented in the current submission.

- **Technically incorrect statements**: There are many hand-wavy and sometimes incorrect statements. First of all, authors should cite the paper "Kiefer et al,  The Weisfeiler-Leman Dimension of Planar Graphs is at most 3, LICS 2017" or its journal version and present the result correctly: The result states that 3-WL is complete on planar graphs (and not just on outerplanar graphs). Moreover, authors also confuse the dimension counts of WL: "...any GNN which matches the expressivity of 3-WL, such as 3-IGN (Maron et al., 2019) or 3-GNN (Morris et al., 2019), is capable of solving our main goal of distinguishing all outerplanar graphs." This is incorrect since Kiefer et al's result uses the classical WL algorithm also referred as the folklore 3-WL, so neither 3-IGN nor 3-GNNs have folklore 3-WL power. It is also open whether folklore 2-WL would suffice for planar isomorphism testing.

- **Significance and novelty**: The paper's technical contribution is limited. Leveraging 1-WL result in combination with CAT is interesting but relatively straightforward. My biggest concern is that there are complete neural models on planar graphs, which are also very scalable. The other technical contributions amount to arguing about shortening the propagation distance in the graph for better information flow. I find this analysis somewhat weak, because this effect can be trivially achieved by adding a virtual node or alike. It appears tangential to the study.

- **Baselines**: I understand that the authors would like to convey the idea of empowering existing GNN models with CAT, but I still think the comparison should be broader when it comes to baseline models.There are many models, including e.g. CIN which achieves very strong results on ZINC. There is also the question of whether this method is applicable to a broader class of models.

**Questions:**

Please refer to my review.

---

> ### Author Response · Authors · 2023-11-15
>
> Thank you very much for your review.
>
> **Weakness Scholarship:**
> We address [1] in an answer to all reviewers, but we would be happy to address any further comments you may have. We would like to emphasize that while we focus on the smaller group of outerplanar graphs, our pre-processing works in linear time.
>
> **Weakness Technically Incorrect Statements:**
> > Technically incorrect statements: There are many hand-wavy and sometimes incorrect statements.
>
> We are happy to clarify any issues and provide more details if some statements were unclear. Please point us to additional statements you find hand-wavy or even incorrect, besides the one regarding the WL-dimension.
>
> Regarding the latter statement, we have to disagree. There seems to be a small misunderstanding between the WL-dimension of outerplanar graphs and the WL-dimension of planar graphs. We purposely cited Kiefer's PhD thesis [2] instead of [3], as only her thesis contains a result specifically for outerplanar graphs. In particular, Corollary $7.40$ [2] states that the WL-dimension of outerplanar graphs is $2$. Note that what Kiefer calls WL is in the GNN literature typically called FWL (folklore WL). This means that $2$-FWL is necessary (i.e., $1$-WL / color refinement is not enough) and sufficient ($3$-FWL is not required) for outerplanar graphs. Furthermore, 2-FWL is equivalent to what the GNN community calls 3-WL. By the known equivalence of, e.g., $3$-WL-GNNs / $3$-IGN to $3$-WL and hence $2$-FWL, we see that the two mentioned GNNs are the $k$-IGN / $k$-WL-GNNs variants with minimum $k$ that one would need to distinguish all outerplanar graphs. Note that they have a runtime of roughly $\mathcal{O}(n^3)$.
>
> By contrast, [3] shows that the WL-dimension of planar graphs is at most $3$ (and it is still open whether it could be $2$). Hence $3$-FWL and $4$-WL are sufficient to distinguish all planar graphs. By the previous explanation, this would require (as long as it remains open whether the WL-dimension is $2$ or $3$) to use $4$-IGNs / $4$-WL-GNNs, with an impractical runtime of roughly $\mathcal{O}(n^4)$.
>
> **Weakness Significance and Novelty:**
>
> > My biggest concern is that there are complete neural models on planar graphs, which are also very scalable.
>
> To the best of our knowledge, there are no linear-time complete models for outerplanar graphs except $\text{CAT}$. PlanE requires quadratic time preprocessing (see global response).
>
> > The other technical contributions amount to arguing about shortening the propagation distance in the graph for better information flow. I find this analysis somewhat weak, because this effect can be trivially achieved by adding a virtual node or alike. It appears tangential to the study.
>
> This can indeed be trivially achieved with a virtual node or with other graph rewiring methods. However, the main goal of $\text{CAT}$ was to improve expressivity. As $\text{CAT}$ adds nodes and edges to the graph, we wanted to analyze how this affects other graph properties important for learning.  As such, we believe that showing that $\text{CAT}$ does not increase the diameter arbitrarily (and actually decreases diameter and average effective resistance in practice) is a useful result. After all, our analysis shows that it is **not** necessary to employ methods like virtual nodes in conjunction with $\text{CAT}$.
>
> **Weakness Baselines:**
> We are in the process of conducting additional experiments (see global response).
>
> [1] Dimitrov et al., PlanE: Representation Learning over Planar Graphs. NeurIPS 2023.
> [2] Sandra Kiefer, Power and limits of the Weisfeiler-Leman algorithm, PhD Thesis at RWTH Aachen, 2020.
> [3] Kiefer et al., The Weisfeiler-Leman Dimension of Planar Graphs is at most 3, LICS 2017

---

> > ### Comment · Reviewer_geBr · 2023-11-21
> > **Thanks for the clarifications**
> >
> > I thank the authors for their work during the rebuttal and for the clarifications:
> >
> > - Regarding 2WL: I understand the explanation provided by the authors but the confusion is natural given the writing of the paper. I recommend they explicitly cite the 3WL bound for planar graphs, and then state 2WL bound for outerplanar graphs to put things in better perspective. Additionally, they should make it clear which hierarchy they are referring to especially when they refer to the $k$ dimensional version of the algorithm.
> >
> > - Regarding novelty, significance, and scholarship: All my concerns remain in terms of these aspects. I appreciate the linear runtime, but this is easy to achieve if we focus on such a small class of graphs. PlanE achieves this for the full class of planar graphs: authors state quadratic time preprocessing as a limitation, but it is clear that this  is due to the difficulty of handling all the triconnected components (which is necessary if one wants to capture the full class). If one is happy with a smaller class of planar graphs, then it is quite easy to design linear-time preprocessing as well. In fact, Block cut trees and even SPQR trees can be constructed in linear time.
> >
> > Overall I think this is a nice trick to enhance GNNs, but it is very incremental in my opinion.

---

> > > ### Author Response · Authors · 2023-11-21
> > >
> > > Dear reviewer,
> > >
> > > thank you for your reply.
> > >
> > > **Regarding Comparisons to PlanE:**
> > > We would like to address your concern that our work is incremental compared to PlanE. Besides the discussed different runtime complexities, the major difference is that $\text{CAT}$ is a preprocessing / graph transformation and PlanE is a custom neural architecture. Graph transformations have the benefit that they can easily be combined with any GNN (independent of the deep learning framework) [1, 2].
> > >
> > > Furthermore, we have analyzed the differences in expressivity between $\text{CAT}$+MPNN and PlanE more closely and have found that in fact $\text{CAT}$+MPNN and PlanE are **incomparable with resepect to expressivity**. This follows from these two observations:
> > > 1. You can find a pair of planar non-isomorphic graphs which can be distinguished by PlanE  (as PlanE is complete on planar graphs) but not by $\text{CAT}$+MPNN (after transforming the graphs with $\text{CAT}$ they remain indistinguishable by WL) [here](https://anonymous.4open.science/r/OuterplanarICLR_Examples-6580/PlanE_beats_CAT.png).
> > > 2. You can find a pair of non-isomorphic non-planar graphs which cannot be distinguished by WL [here](https://anonymous.4open.science/r/OuterplanarICLR_Examples-6580/CAT_beats_PlanE.png). As the graphs are non-planar PlanE cannot be applied to them and thus it cannot distinguish them. Conversely, $\text{CAT}$+MPNN can distinguish them as both graphs have non-isomorphic outerplanar biconnected components.
> > >
> > > **Regarding $2$-WL vs $3$-WL:**
> > > We agree and will improve the exposition of this paragraph in the final version.
> > >
> > > **Regarding the Number and Importance of Outerplanar Graphs:**
> > > Per our Table 1, outerplanar graphs make up 92-98% of graphs in common (pharmaceutical) molecular benchmark datasets. Thus, outerplanar graphs form a large fraction of real-world graphs of high relevance to important application.
> > >
> > > **Regarding Adapting PlanE to Outerplanar Graphs:**
> > > > [...] it is quite easy to design linear-time preprocessing as well. In fact, Block cut trees and even SPQR trees can be constructed in linear time.
> > >
> > > While your approach sounds plausible, we are not aware of any publication achieving achieving maximal expressivity on outerplanar graphs in linear time.
> > >
> > > **ICLR Reviewer Guidelines on Contemporaneous Work:**
> > > Independently of the points above, we would like to point out the [ICLR reviewer guidelines](https://iclr.cc/Conferences/2024/ReviewerGuide) which state:
> > > > Q: Are authors expected to cite and compare with very recent work?  What about non peer-reviewed (e.g., ArXiv) papers? (updated on 7 November 2022)
> > >  A: We consider papers contemporaneous if they are published  (available in online proceedings) within the last four months. That  means,  since our full paper deadline is September 28, if a paper was published  (i.e., at a peer-reviewed venue) on or after May 28, 2023,  authors are not required to compare their own work to that paper. [...]
> > >
> > > PlanE appeared on arXiv within the four month window (3 July 2023) and has only been published at NeurIPS after the ICLR submission deadline.
> > >
> > > [1] Veličković, [Message Passing All The Way Up](https://openreview.net/forum?id=Bc8GiEZkTe5), GTRL @ ICLR, 2022
> > > [2] Jogl et al., [Expressivity-Preserving GNN Simulation](https://openreview.net/pdf?id=ytTfonl9Wd), NeurIPS, 2023

---

> > > > ### Comment · Reviewer_geBr · 2023-11-22
> > > >
> > > > Thank you for your response. I respectfully disagree with the authors: I find the proposed approach incremental (regardless of PlanE and other works), because outerplanar graphs is a small class and it is very easy to design linear-time algorithms to strictly subsume the proposed algorithm in this paper. I have already pointed to one way of achieving this, but there are many more. These are very well-known graph structural encodings and are heavily used in the classical graph theory literature. It would have been much more interesting if the proposal was a linear-time algorithm which would apply to a larger class (and had really pushed the limits of linear time encodings).

---

> ### Author Response · Authors · 2023-11-20
>
> Thanks again for the feedback on our submission. We hope we addressed your concerns and are happy to discuss or clarify further. In particular, we discussed PlanE and added a dataset with over 250k molecules (ZINC 250k) in our global reply above.

---

### Official Review · Reviewer_CdS8 · 2023-11-05

**Soundness:** 3 good
**Presentation:** 2 fair
**Contribution:** 3 good
**Rating:** 6
**Confidence:** 3

**Summary:**

This paper investigates the expressive power of graph neural networks (GNNs) for outerplanar graphs. The authors prove that if the cyclic adjacency transform (CAT) of two outerplanar graphs cannot be distinguished by the Weisfeiler-Leman (WL) test, then the two graphs must be isomorphic, which implies that GNNs have enough expressive power to represent properties of outerplanar graphs. Some numerical results are reported and show that the proposed approach is promising.

**Strengths:**

1. This paper theoretically proves that two outerplanar graphs are isomorphic if and only if their CAT cannot be distinguished by the WL test. I think this is the right direction for studying the expressive power of GNNs and the WL test -- one should focus on some specific class of graphs since in general WL test cannot solve the graph isomorphism problem perfectly.
2. The authors give an explicit example of a pair of non-isomorphism outerplanar graphs that cannot be distinguished by the WL test, showing that CAT is necessary. The proposed approach is efficient, in the sense that the CAT of outerplanar graphs can be computed in linear time.
3. Outerplanar graphs are of practical interest in modeling molecules and the reported numerical results look nice.

**Weaknesses:**

1. The presentation can be improved. For example, maybe the authors can consider presenting two molecules in Figure 5 before at the beginning of Section 3. It might be better to explain the intuition of Theorem 1 and Theorem 2 using specific examples (like Figure 5).
2. It might be better if the authors can discuss more about the limitations and the potential of the mathematical techniques  -- For example, does Theorem 2 hold true for general planar graphs, why or why not?

**Questions:**

None.

---

> ### Author Response · Authors · 2023-11-15
>
> Thank you very much for your review and remarks, which we are happy to discuss.
>
> **Weakness 1.** Improve presentation:
>
> We will improve the presentation as suggested and motivate the two theorems better.
>
> **Weakness 2.** Discuss more (theoretical) limitations. Does Theorem 2 hold true for general planar graphs, why or why not?
>
> Currently, we have not investigated whether planar graphs in general can be distinguished by $\text{CAT}$. However, we conjecture that WL+$\text{CAT}$ cannot, since the biconnected components in general planar graphs do not necessarily have a unique Hamiltonian cycle.
> However, $\text{CAT}$ can also be applied to non-outerplanar graphs and potentially increase the expressivity. Additionally, we have first ideas how to generalize $\text{CAT}$ to distinguish all ($k$-outer)planar graphs (see answer to Reviewer DwdD, Weakness 3.).

---

> ### Author Response · Authors · 2023-11-20
>
> Thank you again for your review! We hope that we could resolve all of your questions.

---

> > ### Author Response · Authors · 2023-11-21
> >
> > > It might be better if the authors can discuss more about the limitations and the potential of the mathematical techniques -- For example, does Theorem 2 hold true for general planar graphs, why or why not?
> >
> >
> > We are happy to address your question. While $\text{CAT}$ cannot distinguish all planar graphs, it can distinguish some non-planar graphs. We have linked example figures in the global response.

---

### Official Review · Reviewer_DqdD · 2023-11-09

**Soundness:** 3 good
**Presentation:** 3 good
**Contribution:** 3 good
**Rating:** 5
**Confidence:** 4

**Summary:**

This paper studies the problem of how to distinguish non-isomorphic *outer-planar* graphs using GNNs. The motivation is that: (1) distinguishing non-isomorphic in the general setting is intrinsically hard; (1) the restricted outer-planar graphs is common in various practical settings, so solving this simpler problem is always significant in real-world applications; (3) it is possible to design extremely efficient GNN models to solve outer-planar graph isomorphism, which only have a linear complexity.

The authors thus used theoretical results related to outer-planar graphs to design a *preprocessing step* so that the processed (directed) graphs can be distinguished by 1-WL. They also discussed the property of the processed graphs, such as diameter and resistance distance. The approach is efficient in that the graphs are a linear number of vertices and edges.

Finally, the authors conducted experiments to show the effectiveness of the proposed approach.

**Strengths:**

1. The paper is well-motivated. I appreciate the topic of studying GNN design for outer-planar graphs: although it is somehow restricted, it still covers important practical applications and sounds reasonable to me.
2. The paper is clearly written. Despite the sophisticated theoretical background, the presentation is generally easy to read and organizes well. The notations are consistent, and the presented figures is great and intuitive. I particularly appreciate the counterexamples in this paper, which gives insights into the proposed algorithm.
3. The proposed method is efficient. It has a linear (worst-case) complexity, not only due to using the standard MPNN, but even for the preprocessing step, which contrasts to prior work.

**Weaknesses:**

1. Discussions of related work could be more comprehensive. For example, I found that this paper is highly related to the recent paper [1]. Both works share a common foundation rooted in biconnectivity, specifically concerning biconnected components and block cut trees, as well as a focus on distance metrics, encompassing the shortest path distance and resistance distance. Since the motivation of the two papers is also similar (both pointing out the importance of planar graphs in practice and giving molecular graphs as counterexamples), I feel that giving an in-depth discussion (perhaps within the introductory section of your paper), could further justify the paper's significance and provide a more comprehensive context for the reader.

   Besides, I also found that another recent paper [2] studied a similar topic to your paper but this paper currently did not discuss it. Could you provide some discussions for the similarity and difference between the two works?

2. The proposed CAT transformation is somehow complicated, in particular for the general case (Definition 2). I think it may be beneficial to discuss more about the intuition of Definition 2 in Appendix. Moreover, it introduces a lot of additional nodes and edges and even changes an undirected graph to a directed one. While it has indeed shown that the separation power is improved, significantly changing the structure of the input graph may result in some drawbacks. As another problem, can the approach be extended to node classification?

3. The proposed approach only works for outer-planar graphs. Can the results be generalized to general planar graphs? I think Lemma 1 does not hold in the general setting. Since still about 5% of the graphs are not outer-planar as shown in Table 1, this is perhaps a limitation.

4. While the experimental results showed that an MPNN with CAT can outperform vanilla MPNN, such baselines are actually quite weak. I am not fully convinced to what extent the proposed approach is superior in practice.

Despite the weaknesses and questions, I still appreciate this paper and tend towards giving an acceptance. I will consider increasing the score if the concerns are well-addressed.

Miscellaneous minor issue: in the fourth line in page 8, there is a redundant word "and".


[1] Rethinking the expressive power of GNNs via graph biconnectivity. ICLR 2023.

[2] PlanE: Representation Learning over Planar Graphs. NeurIPS 2023.

**Questions:**

Can you illustrate more about how your approach may/ may not be generalized to $k$-outerplanar graphs?


---

The authors addressed several of my concerns, but it seems that there is indeed a close relation between this paper and PlanE after seeing other reviews, and PlanE can solve the more general planar graph isomorphism with still linear complexity (if not considering the preprocessing step). Considering all these aspects, I thus decided to maintain my score.

---

> ### Author Response · Authors · 2023-11-15
>
> Thank you very much for the review and remarks, which we are happy to discuss.
>
> **Weakness 1.** Related Work:
> Thank you for your remarks regarding [1]. Yes, we are happy to extend our discussion of [1] in the related work in the final version of our submission. Most importantly note that all GNNs discussed in [1] have a time complexity of roughly $\mathcal{O}(n^2)$, while our approach has linear time complexity for both pre-processing and message passing. Additionally, we would like to point out that merely identifying block and cut vertices (and edges etc.) is not enough to distinguish outerplanar graphs. For example, the two non-isomorphic biconnected outerplanar graphs in our Figure 1 (as undirected graphs) have the same block-cut-vertex-trees (actually just one node).
>
> Moreover, it is worth noting that [1] proposes an architecture that relies on a transformer backbone, while $\text{CAT}$ is a graph transformation (i.e., a simple pre-processing), which allows for maximally expressive MPNNs (and any GNN that has at least WL expressivity) on outerplanar graphs.
> We address the relation to the PlanE paper in our answer to all reviewers and will include a discussion thereof in the related work section of our final submission.
>
> **Weakness 2.** Intuition:
> We will add a thorough description of the intuition of $\text{CAT}$ to the appendix and include further examples.
>
> > While it has indeed shown that the separation power is improved, significantly changing the structure of the input graph may result in some drawbacks.
>
> One possible drawback we considered was that $\text{CAT}$ might reduce graph connectivity. That is why we included an empirical analysis (Table 3) plus some theoretical guarantees (Proposition 1). We prove that even in the worst case $\text{CAT}$ never increases the diameter by more than 7. In the empirical analysis we actually observe that connectivity (in terms of average pairwise effective resistance and diameter) decreases in almost all cases. Could you please clarify which other drawbacks are meant here?
>
>
> > Can the approach be extended to node classification?
>
> Yes, there is a straightforward extension of our approach to node classification: after the message passing of the MPNN it suffices to apply a multilayer-perceptron to the node representation of the node we want to classify. This has not impact on the theoretical guarantees of $\text{CAT}$. For the two example graphs from Figure $1$, $1$-WL cannot distinguish the nodes of degree $3$ between the two graphs. With $\text{CAT}$, however, these nodes can be distinguished.

---

> > ### Author Response · Authors · 2023-11-15
> >
> > **Weakness 3 / Question**: "Can you illustrate more about how your approach may/ may not be generalized to $k$-outerplanar graphs?"
> >
> > The main property of outerplanar graphs that we use is that their biconnected components have unique and efficiently computable Hamiltonian cycles. For more general graph families, we would need to decompose the graph into subgraphs with unique Hamiltonian cycles or perhaps iterate over all Hamiltonian cycles. Each of the Hamiltonian subgraphs could then be processed by something similar to $\text{CAT}^*$. How to perform this decomposition is a highly non-trivial question. Moreover, there might be different decompositions of the same graph and so we would have to enumerate all possible decompositions, if we want to remain permutation-invariant.
> >
> > We chose $k$-outerplanar graphs as an example for future work, as the small percentage of non-outerplanar graphs in molecular datasets is known to be $k$-outerplanar for a small $k$ (typically, $2$-$4$, see e.g. [4]). Also note that any planar graph is $k$-outerplanar for some $k$. Furthermore, $k$-outerplanar graphs, once embedded into the plane in a $k$-outerplanar way, allow per definition for a decomposition into the $k$ many ``shells''. Similarly one could inspect the _book embedding_ of a $k$-outerplanar graph, which effectively decomposes the graph into up to $3k$ outerplanar subgraphs (i.e. _pages_) [2]. As for typical molecules $k$ is a very small constant, we expect the number of different such decompositions to be manageable. We regard this as a promising future research direction.
> >
> > **Weakness 4.** Experiments:
> > > While the experimental results showed that an MPNN with CAT can outperform vanilla MPNN, such baselines are actually quite weak. I am not fully convinced to what extent the proposed approach is superior in practice.
> >
> > We will provide additional experiments and comparisons to a stronger baseline (see global response).
> > While MPNNs are no longer considered state of the art, we disagree that they are quite weak. For example, our GIN on MOLHIV outperforms 7 out of 8  models proposed in [3]. Our current experiments indicate that $\text{CAT}$ improves the predictive performance of MPNNs. We expect that $\text{CAT}$ can improve the performance for other GNNs.
> >
> > [1] Zhang et al., Rethinking the expressive power of GNNs via graph biconnectivity. ICLR 2023.
> > [2] Ganley and Heath. "The pagenumber of k-trees is O (k)." Discrete Applied Mathematics 109.3 (2001): 215-221.
> > [3] Bevilacqua et al., Equivariant Subgraph Aggregation Networks. ICLR 2022.
> > [4] Horváth et al., Efficient frequent connected subgraph mining in graphs of bounded tree-width. TCS 411 (2010) 2784–2797

---

> ### Author Response · Authors · 2023-11-20
>
> Thank you again for your review! We hope that we could resolve all of your questions. In particular, we discussed PlanE and added experiments on a larger dataset with over 250k molecules (ZINC 250k) in our global reply above.

---

### Official Review · Reviewer_zczo · 2023-11-09

**Soundness:** 3 good
**Presentation:** 3 good
**Contribution:** 2 fair
**Rating:** 5
**Confidence:** 3

**Summary:**

In this work, the authors studied the isomorphic problem of a special graph family, outerplanar graphs, by using GNNs. Compared to general graphs, most pharmaceutical molecular graphs correspond to this graph family. To this end, the authors proposed an efficient graph transformation approach to enhance the expressiveness of classical MPNNs. Additional theoretical analysis is provided to show that maximum expressivity on outerplanar graphs is achieved with the proposed approach. Empirical experiments are conducted to demonstrate the effectiveness of the proposed approach.

**Strengths:**

1. The targeted problem is of great practical significance. As pointed out in this paper, most pharmaceutical molecules correspond to outerplanar graphs, which play important roles in real-world applications in chemistry and biology.

2. The proposed CAT approach is both efficient (in linear time) and theoretically sound.

**Weaknesses:**

1. Lacking clarifications and discussions of related work. In [1], the authors developed a framework for the whole planar graphs class, which seems very relevant to this work. It is highly recommended to (1) clarify the novelty and originality of this work against [1]; (2) add discussions on the relations, scopes and any other aspects of these two works for improved quality.

2. The empirical evaluation needs to be further improved. Although the authors provided results on several benchmark datasets, I still think the evaluation does not meet the bar of this conference:
    - In Table 4, CAT+GAT consistently underperforms GAT on most datasets. Such a degrade performance brought by CAT is strange compared to that in GIN/GCN. Could you provide further explanation on this phenomenon?
    - The experimented baselines are limited. In Table 4, only GCN,GIN and GAT are tested. In recent years, there exist advanced GNN variants with linear complexity. To better verify the generality of the proposed approach, it is highly recommended to conduct more experiments on other GNNs.
    - The scale of chosen benchmarks is limited. From Table 1, we can see that there also exist large-scale molecule datasets (>100k) suitable to verify the approach for outplanar graphs. It is highly recommended to conduct more experiments on more large-scale datasets to see whether the proposed approach can consistently bring gains in this setting.

[1] Dimitrov, R., Zhao, Z., Abboud, R., & Ceylan, I. I. (2023). PlanE: Representation Learning over Planar Graphs. arXiv preprint arXiv:2307.01180.

**Questions:**

Please refer to the Weaknesses section

---

> ### Author Response · Authors · 2023-11-15
>
> Thank you very much for the review. We address some of your remarks in our global reply to all reviewers, but we would be very happy to discuss any additional comments you may have.
>
> **Weakness 1.** (Theoretical) comparison to PlanE:
> We address the PlanE paper in detail in our answer to all reviewers and will include it in the related work in the final submission.
>
> **Weakness 2.** More experiments:
> We will include further experiments in the final version, please also refer to the answer to all reviewers regarding additional experiments.
>
> > In Table 4, GAT+$\text{CAT}$ consistently underperforms GAT on most datasets. Such a degrade performance brought by $\text{CAT}$ is strange compared to that in GIN/GCN. Could you provide further explanation on this phenomenon?
>
> We are also surprised of the performance of GAT+$\text{CAT}$ on these datasets. Note however, that we could significantly boost the performance of GAT on ZINC. In particular, commonly reported test MAE values for ZINC are around $0.475$ [1]. By contrast we achieve a performance of $0.375$ for GAT alone and of $0.201$ for GAT+$\text{CAT}$. We leave further investigation for future work.
>
> > In recent years, there exist advanced GNN variants with linear complexity.
>
> We would be happy to discuss these GNNs in our work. However, we are not aware of such GNNs. Most recent GNNs that we are aware of, in particular ones that are more expressive than MPNNs  (e.g. ESAN, CIN, subgraph based GNNs, ...), have superlinear complexity for pre-processing and/or message passing. Could you please tell us what you had in mind?
>
> > The scale of chosen benchmarks is limited. From Table 1, we can see that there also exist large-scale molecule datasets (>100k) suitable to verify the approach for outplanar graphs. It is highly recommended to conduct more experiments on more large-scale datasets to see whether the proposed approach can consistently bring gains in this setting.
>
> We are in the process of implementing further experiments on this (see global response) and will let you know once we have results.
>
> [1] Dwivedi et al., Benchmarking Graph Neural Networks, JMLR 2022.

---

> ### Author Response · Authors · 2023-11-20
>
> Thank you again for your review! We hope that we could resolve all of your questions. In particular, we discussed PlanE and added experiments on a larger dataset with over 250k molecules (ZINC 250k) in our global reply above.

---

### Author Response · Authors · 2023-11-15

Dear reviewers,
Thank you very much for the feedback on our submission. We will address some general points here and also provide answers to the reviewers individually below.

## PlanE
Thank you for bringing [1] to our attention. We consider [1] contemporaneous to our submission (NeurIPS papers were released three weeks ago). We will, however, include [1] in the related work section of the final version of our submission and discuss differences in detail. [1] introduces a graph learning algorithm which is complete for planar graphs, while we focus on outerplanar graphs. [1] builds upon the algorithm KHC [2] to canonically label planar graphs, while we make use of the fact that Hamiltonian cycles are unique in biconnected outerplanar graphs [3].

**Advantages of our approach**:
* Our algorithm $\text{CAT}$ is a graph transformation (i.e., a simple pre-processing of the graph), and thus $\text{CAT}$ can be easily combined with existing GNN architectures without additional modifications. We focus on outerplanar graphs, as most molecule datasets consist of outerplanar graphs.
* Time complexity: [1] needs a quadratic pre-processing time of $O(|V|^2)$ in the number of nodes, while our graph transformation runs in **linear** time for pre-processing and message passing.
* Our algorithm can also be applied to any non-outerplanar graphs, where its runtime remains linear. In fact this is what we do in our experiments. If the non-outerplanar graphs have any outerplanar biconnected components, $\text{CAT}$ can increase the expressivity. By contrast, PlanE cannot be applied to non-planar graphs and the authors mention that such a generalization would be non-trivial.
* We investigated the impact of our transformation on graph connectivity, and experimentally confirm that the effective resistance decreases. This type of analysis is not present in [1].

Overall we think it is exciting that there is growing interest in devising maximally expressive GNNs for practically important graph families such as (outer)-planar graphs.

## Further experiments:
We try our best to implement and run further experiments. We intend to include an additional state-of-the-art GNN (both as a baseline and combined with $\text{CAT}$) and a larger dataset. If we manage in the limited time of the rebuttal, we will report the results here.

[1] Dimitrov et al., PlanE: Representation Learning over Planar Graphs. NeurIPS 2023.
[2] Kukluk et al. Algorithm and experiments in testing planar graphs for isomorphism. Journal of Graph Algorithms and Applications, 8(3):313–356, 2004.
[3] Colbourn and Booth. Linear time automorphism algorithms for trees, interval graphs, and planar graphs. SIAM Journal on Computing, 10(1):203–225, 1981

---

### Author Response · Authors · 2023-11-20
**Additional Experiments on ZINC 250k**

We have performed additional experiments on the full ZINC dataset (250k graphs) to address the comments about the limited dataset size. Due to the time constraints, we were unable to perform experiments with additional GNN architectures. For experiments on ZINC 250k, we use the same setup as for the subset ZINC experiments (with smaller hyperparameter grids) and evaluate the final model 5 times. The results are as follows:
- GIN: $0.033 \pm 0.003$ mae
- GIN + $\text{CAT}$:  $0.034 \pm 0.003$ mae
- GCN: $0.067 \pm 0.005 $ mae
- GCN + $\text{CAT}$: $0.034 \pm 0.003$ mae
- GAT: $0.103 \pm 0.004$ mae
- GAT + $\text{CAT}$: $0.046 \pm 0.004$ mae

On this dataset,  $\text{CAT}$ reduces the error of GCN and GAT by 50%. This indicates, that the improvements by $\text{CAT}$ are even stronger for large datasets. Note that this is near SotA performance as reported by e.g. [1]: CIN (small) and HIMP both get outperformed by GIN, GIN + $\text{CAT}$ and  GCN + $\text{CAT}$. We hope that this addresses the concerns about larger datasets and weak baselines.

[1] Bodnar et al., Weisfeiler and Lehman Go Cellular: CW Networks, NeurIPS 2021

---

### Author Response · Authors · 2023-11-21
**On incomparable expressivity of CAT and PlanE**

We have analyzed the differences in expressivity between $\text{CAT}$+MPNN and PlanE more closely and have found that in fact $\text{CAT}$+MPNN and PlanE are **incomparable with resepect to expressivity**. This follows from these two observations:
1. You can find a pair of planar non-isomorphic graphs which can be distinguished by PlanE  (as PlanE is complete on planar graphs) but not by $\text{CAT}$+MPNN (after transforming the graphs with $\text{CAT}$ they remain indistinguishable by WL) [here](https://anonymous.4open.science/r/OuterplanarICLR_Examples-6580/PlanE_beats_CAT.png).
2. You can find a pair of non-isomorphic non-planar graphs which cannot be distinguished by WL [here](https://anonymous.4open.science/r/OuterplanarICLR_Examples-6580/CAT_beats_PlanE.png). As the graphs are non-planar PlanE cannot be applied to them and thus it cannot distinguish them. Conversely, $\text{CAT}$+MPNN can distinguish them as both graphs have non-isomorphic outerplanar biconnected components.

---

### Meta-Review · Area_Chair_MZZc · 2023-12-06

**Metareview:**

This work proposes an efficient graph transformation that enables the Weisfeiler-Leman (WL) test and message-passing graph neural networks (MPNNs) to be maximally expressive on outerplanar graphs. With this transformation, authors show that outerplanar graphs can be fully distinguished in linear time. Theoretical results are backed up with empirical experiments.

The reviewers raise two concerns that influence the final decision. First, two reviewers pointed out that this work is similar to a recently published work, "PlanE: Representation Learning over Planar Graphs". The two works contain significant overlaps. Second, three reviewers found that the baselines used in the experiments were weak. The author should choose recent works as baselines and conduct experiments on large-scale datasets. Given those unaddressed concerns, I recommend rejection following the majority and hope the author can address those problems and submit this work to a future venue.

**Justification For Why Not Higher Score:**

See the metareview

**Justification For Why Not Lower Score:**

N/A

---

### Decision · Program_Chairs · 2024-01-16

Reject